# Substitutability and Complementarity of Municipal Electric Bike Sharing Systems against Other Forms of Urban Transport

**Michał Suchanek [1],\*, Aleksander Jagiełło [1] and Justyna Suchanek [2]**

[1] Faculty of Economics, University of Gdansk, Ul. Armii Krajowej 119/121, 81-824 Sopot, Poland; aleksander.jagiello@ug.edu.pl

[2] Independent Department of EU Projects and Mobility Management, Ul. 10 Lutego 33, 81-364 Gdynia, Poland; j.suchanek@gdynia.pl

\* Correspondence: michal.suchanek@ug.edu.pl

**Abstract:** The current quantitative and qualitative development of bike-sharing systems worldwide involves particular implications regarding the level of sustainability of urban development and city residents' quality of life. To make these implications as large as possible as well as the most positive, it is essential that the people who use municipal bikes on a regular basis to the largest extent possible abandon car travel at the same time. Thanks to their operational characteristics, electric bikes should enable meeting the transport needs of a wider group of city residents compared with traditional bicycles. The main aim of this study was therefore to check whether the municipal electric bike system (MEVO) in Gdańsk-Gdynia-Sopot metropolitan area of Poland lived up to the hopes placed upon it by policymakers. Therefore, the article tests the hypothesis indicating that the municipal electric bike systems constitute a substitutable form of transportation against passenger cars to a larger extent than against collective urban transport and walking trips. The analysis was performed based on the results of primary studies conducted among the users of MEVO. The data show that the MEVO was a substitutable form of transportation against collective transport and walking trips to a larger extent than against passenger cars. Through logistic regression analysis, the variables concerning the probability of replacing car trips by MEVO bicycles were determined. Among the analyzed variables, the following turned out to be statistically significant: age, the number of people in the household, the number of cars in the household, the distance from work, and gender. The results therefore indicate that substituting in favor of electro bikes was more probable for younger people with fewer people in the household and a distance to travel below 3 km, whereas it was less probable for people with more cars in the household or traveling a distance longer than 10 km. Additionally, females were more likely to choose the bike system.

**Keywords:** municipal electric bike system; electric bike-sharing; sustainable urban transport; travel behavior

## 1. Introduction

The current urban transport policies focus mostly on the external costs of transport, such as the ones related to air pollution, noise, road congestion, and road accidents. These costs are, to a large extent, a consequence of the dominant share of individual transport, mostly car transport, in the city residents' municipal travels. The political decisions that are made indicate the intention to reduce the share of passenger cars in daily travels. However, it requires providing city residents with an attractive alternative. Fulfilling the city residents' transport needs with the use of bicycles matches the concept of sustainable urban development, since travel by bicycle, apart from walking trips, constitutes the most environmentally friendly mode of transportation of the city residents. To increase the share of bike travels in the modal split of particular cities worldwide, bike-sharing systems have been introduced.

In the Gdańsk-Gdynia-Sopot metropolitan area, the bike-sharing system (called MEVO) was introduced in 2019. In 2019, there were 96 shared bike systems in operation in Poland. Access to bike-sharing services operating in 108 cities had 2.6 million registered users [1]. MEVO was the first bike-sharing system in Poland using only electrical bicycles (with electric power-assisted steering). The main reason for the use of electric bicycles in the MEVO system rather than traditional bicycles was the large differences in the ground levels in the Gdańsk-Gdynia-Sopot metropolitan area. Electric bicycles were therefore intended to enable almost everyone to use the bike-sharing system.

The introduction of the MEVO system into operation was accompanied by the hope of reduced car trips and an increased share of public transport [2]. The increase in the share of public transport was to result from the use of city bikes by residents for the first and last mile of their journeys. Furthermore, the reference literature indicates numerous advantages to fulfilling transport needs with the use of bikes, and consequently, bike-sharing systems, the advantages of which include the following [3–11]:

- Reduced consumption of fuel necessary for meeting the city residents' transport needs;
- Reduced congestion;
- Reduced level of noise and pollution emitted by fuel-powered vehicles;
- Complementing the offer of an urban transport system;
- Supporting park-and-ride systems;
- Offering the "last mile" solution while traveling to places where transportation is limited or prohibited;
- Improved transport availability;
- Increased tourist attractiveness of the city;
- Improved physical and psychological health of the city residents;
- Increased possibilities to meet the city residents' transport needs;
- Increased social acceptance for the introduction of rigorous transport policy toward passenger cars (e.g., reduced number of parking places).

To make it possible for bike-sharing systems to have a positive effect to the largest extent possible through their advantages on the city residents' quality of life, it is preferable for the users of bike-sharing systems to abandon using passenger cars for urban travels. Therefore, it is preferable for municipal bikes to become the substitute for passenger cars to the largest extent possible (particularly owned cars). To make this possible, it is necessary to facilitate not only the bike renting process itself but also traveling by bike. One of the feasible solutions involves using electric bicycles in the bike-sharing systems. Thanks to such a solution, bike travel requires less physical effort, and consequently, it is possible for a larger group of potential users. Definitely less favorable is a situation where the municipal bike users abandon walking trips or traveling by public transport in favor of municipal bikes.

The main objective of this study is to verify whether municipal electric bike systems constitute a substitutable form of transportation against passenger cars. If that is not the case and they substitute public transport instead, then they do not contribute to sustainable urban development to a large extent. Given the foregoing, this article verifies the following hypothesis: municipal electric bike systems constitute a substitutable form of transportation against passenger cars to a larger extent than against collective public transport and walking trips. This study thus shows the real effects of implementing an electric bike-sharing system in the largest metropolitan area in northern Poland and analyzes whether the effects of the implementation of the MEVO system are consistent with the expectations of policymakers. Based on the obtained data, the factors affecting the residents' willingness to substitute car travel with municipal bike travel were analyzed as well.

## 2. Literature Review

### 2.1. History of Bike Sharing—From "White" to Electric Bikes

Bike-sharing systems are based on the idea of sharing economy. Since the concept of sharing economy is only a recent concept in the economic sciences (one of the first persons who used this term in 2008 was Prof. L. Lessig of Harvard Law School [12]), we have not yet observed one commonly applied definition of this term. The sharing economy is most frequently defined from the angle of its usability, as providing consumers with temporary access to unused physical resources is usually performed in return for financial means [13].

The concept of municipal bikes was introduced for the first time in the 1960s [14]. Over the years, the concept evolved from the so-called "white bicycles" into the currently introduced fourth generation municipal bikes. The first attempts to introduce bike sharing should be considered a failure [15]. Even though the bike's gratuitousness and availability was supposed to be its best advantage, quickly (i.e., only within several days), the bikes operating within these systems were damaged by vandals, stolen by thieves, or confiscated by the police [16,17]. In response to the observed problems with the first-generation municipal bike systems, attempts were made to introduce second-generation systems. These attempts were first made in the first half of the 1990s, mainly in Denmark [16]. These bikes were specially adjusted for intensive use, and as a result, they were less prone to accidental damage resulting from normal use. The greatest breakthrough involved applying docking stations, which enabled renting a bike against a deposit [17]. The problem affecting the second-generation systems included the still too significant anonymity of the bike users. This anonymity led to acts of vandalism and bike theft incidents. The third-generation municipal bike systems were successful mainly thanks to the development of technology, which was observed in the second half of the 1990s and later. The drawback of the third-generation municipal bike systems compared with subsequent generations is their dependence on docking stations. This means that the user of a third-generation municipal bike is forced to rent it from and return it to one of the docking stations located in a particular city. The possibility to start and finish the ride outside the docking station in any location (within the area of system operation) is provided by fourth-generation bike-sharing systems. Moreover, they still feature all the advantages of the previous generations' systems. The drawback of the fourth-generation systems refers to the need to reposition the bikes, which was more broadly presented in [18–21].

Each subsequent generation contributed to changing the operational scheme of the municipal bike system. In the case of bike-sharing systems with stations, their success depends on the solution of three optimization problems: the number and locations of stations, the capacity of the stations, and the bicycle distribution [22]. The operational schemes of subsequent generations of municipal bikes from the user perspective are presented in Figure 1. This shows that the subsequent generations of bike-sharing systems not only reduced the bike renting process but also added more elements into the system. Nevertheless, thanks to the application of IT solutions (that allow, for example, to find the accurate location of a vacant bike, pay automatically, and register in the system online), the service of fourth-generation municipal bikes should create no barriers to their use, whereas people who do not have a smartphone are partially excluded from using fourth-generation municipal bikes. In Poland in 2018, 63% of people had a smartphone (the median for the developed countries totaled 76%) [23]. Fourth-generation municipal bike systems are the first to use not only conventional but also electric bikes.

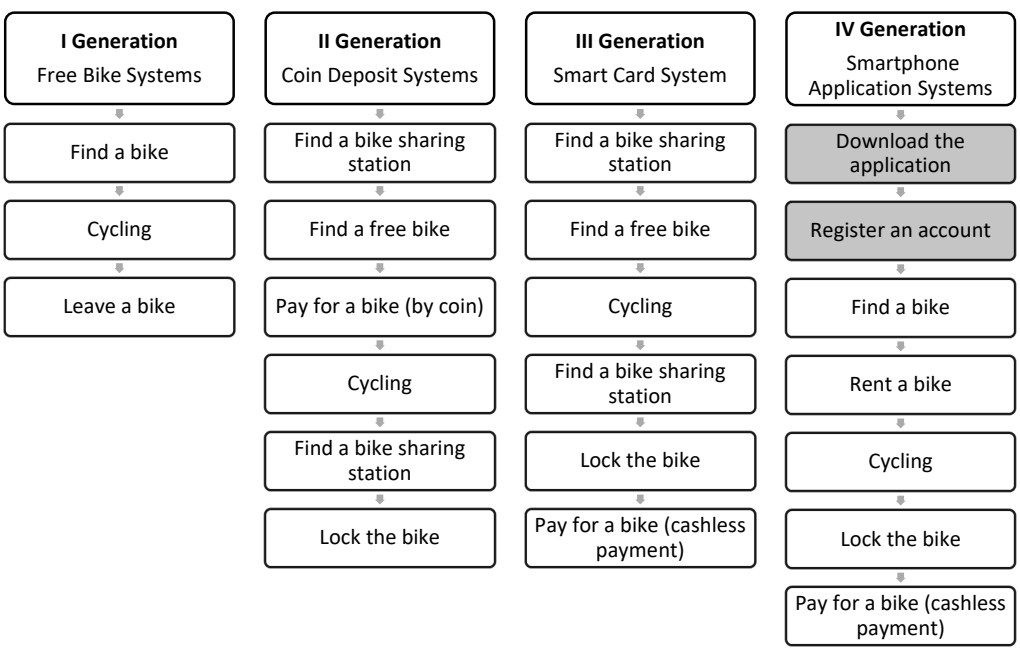

**Figure 1.** Operational schemes of municipal bikes of different generations from the user perspective. Source: own elaboration. Explanatory note: gray elements refer to stages the user is required to perform only once.

### 2.2. Place of Bike Sharing in the Urban Transport System

Between the systems of municipal bikes and other forms of urban transport, we can observe one of the three relations. Municipal bikes may be complementary, substitutable, or unrelated to the other forms of transportation [24]. Some researchers studying this issue analyzed the relations within the municipal bike systems from the angle of their complementarity and substitutability against passenger cars and public transport [25,26]. The relations between municipal bike systems and public transport are particularly complicated since, in such case, we may observe both substitutability and complementarity against public transport.

The complementarity between municipal bike systems and public transport results from the possibility to access the public transport station (first mile) by bike and travel the "last mile" by bike from the station to the destination [27]. Since public transport terminals constitute significant traffic generators, we can observe located in their vicinity the stations of bike-sharing systems [17,28]. Studies show that variables related to public transport (e.g., the distance between the municipal bike stations and public transport terminals or the number of passengers of a particular public transport terminal) affect, in a statistically significant manner, the level of demand for bike-sharing system services [29,30]. Therefore, they constitute a good predictor of the volume of demand for the services of particular stations of municipal bike systems.

On the other hand, municipal bike systems constitute the substitutable means of transportation against public transport, particularly for people who do not own passenger cars. Studies have proven that the substitutability of bicycles against public transport is particularly high in the case of short travels with distances of up to 5 km [31–34]. The use of electric bikes in municipal bike systems extends the average travel distance, which involves the substitution of public transport [35]. Numerous studies have proven that the average distance of electric bike travel is longer by roughly 50% than travel by traditional bikes [36,37].

The situation where the municipal bike system substitutes public transport, namely the situation where the city residents abandon using public transport services in favor of

bike-sharing services, involves two consequences. First, the decrease in the number of public transport users will result in a decrease in the public transport fee rate. The public transport service fee rate shows to what extent the total costs of services are covered by the revenue from ticket sales. Therefore, a decrease in the public transport service fee rate means that public transport must be subsidized to a larger extent from the local authority resources. At present, within the area where the MEVO municipal bike system was operational, there are two main public transport operators: ZKM in Gdynia and ZTM Gdańsk. In the case of these two operators, the value of the fee rate does not exceed 50% (ZKM in Gdynia: 43.8%; ZTM Gdańsk: 48%). This means that more than half of the public transport operational costs within the network managed by the above-mentioned operators is subsidized from the local authority budgets. Further outflow of public transport passengers and a decrease in the fee rate may lead to a situation where it is necessary to reduce the public transport offered. Such a situation may lead to a decrease in the attractiveness of the offered public transport and consequently to a further decrease in the number of public transport users. Therefore, fulfilling transport needs with the use of municipal bikes and abandoning the fulfilment of those needs through public transport may lead to increasing the phenomenon defined in the reference literature as "the public transport vicious circle". Despite the fact that "the public transport vicious circle" most frequently occurs during the debate on the place and importance of passenger cars in the municipal transport system, substituting public transport with municipal bikes may lead to increasing this phenomenon, since a decrease in the number of public transport passengers resulting in a decrease in the quality of public transport offered or an increase in ticket prices may lead to the situation where people using public transport services up to this point to meet their transport needs abandon these services and swap for passenger cars (referring in particular to people disinterested in meeting their transport needs with the use of municipal bike systems). Such a situation is currently observed, inter alia, in Latin American cities [38]. It is worth noting that due to the car being a generally less sustainable mode of transport than public transport, if a bike-sharing system substitutes public transport rather than the car, then the overall positive effect in terms of sustainability might be far less significant than if it substitutes the car.

*2.3. Differences between Conventional and Electric Bikes in the Context of Bike-Sharing Services*

Studies have proven that there are certain factors affecting the intention to make use of bicycles by the city residents. The frequency of using municipal bike systems is affected, for example, by age, BMI (body mass index, one of the main health-related indicators), and the users' physical fitness [39–45]. In this context, it should be noted that the analyzed MEVO municipal bike system included only bicycles equipped with electric power-assisted steering (pedelec models, or pedal electric cycles). In these bikes, in order to activate the power-assisting system, the user must push on the pedals. Since the bicycles were equipped with electric power-assisted steering, it is likely that the users of the system also included people who would not decide to use the traditional city bikes (those without the electric power-assisted steering). The people who considered electric power-assisted steering as particularly important included the obese, the elderly, and people with poor physical fitness or health problems (e.g., knee problems, arthritis, asthma, and back pain) [46–48]. These are people who prefer using passenger cars to meet their transport needs. Taking into account the specificity of the MEVO municipal bike system, it can be assumed that the system substitutes, to a larger extent, travel by passenger car compared with the traditional municipal bike systems (with no electric power-assisted steering). It is important in so far as under the review of a conducted survey, it can be concluded that the municipal bike systems using traditional bikes (with no electric power-assisted steering) substitute the sustainable forms of transportation (e.g., public transport, walking trips, and travel by private bicycle) to a larger extent than travel by passenger car [15,24,49–51]. However, it must be emphasized that some analyses (in particular these related to mid-sized cities) indicate the reverse dependency [52]. The fact that electric

bikes can replace passenger cars in urban travels to a larger extent than public transport is confirmed by the results of studies conducted among the bike owners [36,53,54]. It should be noted that these studies included people who owned electric bikes and therefore consciously made the decision to change their transport behavior. In his work, J. Arendsen emphasizes that there are particular (often significant) barriers to changing city residents' behavior and transport preferences [55]. These barriers are usually overcome as a result of serious life changes (e.g., change of work, place of residence, or health condition). Regarding the city residents' strong habits of fulfilling their transport needs, and in particular strong attachment to owning passenger cars, it is justifiable to conclude that the shift from passenger cars to bicycles in daily urban travel should be easier in the case of electric bikes. Thanks to the electric power-assisted steering, such travel is easier, more convenient, and faster, which reduces the discrepancy in the perceived difficulty between traveling by bike and passenger car [56,57].

## 3. Methodology and Data Source

In order to verify the above hypothesis and achieve the main goal of the article, the transport behavior of MEVO municipal bike system users was analyzed. The analysis was conducted based on the method of individual, direct interviews with the use of an original questionnaire. The interviews were conducted at the turn of August and September 2019 (i.e., in the months when the weather in Poland does not hinder the use of bicycles to meet transport needs and also pre-COVID-19). The survey sample included 500 respondents. Before the survey was taken, there had been a pilotage of the questionnaire, which included 12 local experts discussing the questionnaire. The experts included 6 researchers, 2 bike activists, and 4 employees of the town halls specializing in the public transport domain. The questionnaire consisted of 18 questions, including one screening question. The given screening question allowed for the omission of respondents who did not live in the area served by the MEVO system (e.g., tourists). The pool of the final respondents included the municipal bike system's actual users who left or rented municipal bikes at one of six stations located in the three main cities (Gdynia, Gdańsk, and Sopot) where the MEVO municipal bike system subject to analysis was operational. The stations where the survey was conducted included six stations which were all relatively close to public station stops (bus, trolley, or municipal rail):

− Galeria Bałtycka (station No. 11365);
− Olivia Business Centre (station No. 11358);
− Gdynia Główna (station No. 12000);
− Gdynia City Museum (station No. 12053);
− Skwer Kuracyjny Sopot (station No. 10100);
− Bohaterów Monte Cassino (station No. 10124).

It should be emphasized that MEVO functioned as a point-area system (i.e., the return of the bicycle can take place at the station but also by leaving the bike anywhere in the designated area after paying an additional fee). The stations surveyed were selected so that their operational scope covered some of the main traffic generators located within the area covered by the MEVO system of various specificity (e.g., traffic generators related to work, recreation, or shopping). Thereby, the survey covered people who used the municipal bike system to differentiate the goals (meeting the transport needs related to diversified destinations). According to the data of the system operator, these stations were the most popular stations of the MEVO system in individual cities [58]. It should also be noted that an equal number of interviews were conducted at each station.

Upon conducting the survey, the MEVO system comprised 1224 bicycles equipped with electric power-assisted steering. Ultimately, the system was supposed to include 4080 electric bicycles, which would make it the largest bike-sharing system of this type in Poland and Europe [59]. The lack of experience in the exploitation of electric bikes by the system operator (Nextbike, the European bike-sharing market leader) resulted in the

suspension of the whole system after seven months of operation. A particularly big problem for the system operator was the charging of the batteries in the bicycles and the redistribution of the charged vehicles. Despite the attempts made, the system was not brought back into operation due to an ongoing legal dispute.

Table 1 presents the most important characteristics of the survey sample. The respondents comprised a similar number of women and men. The gender parity with slightly more women among the MEVO municipal bike users complied with the gender structure of the inhabitants of Poland and was similar to the parity among cyclists in the European conditions [60]. The majority of respondents had a driver's license and a passenger car in their household. Therefore, they could make a decision to fulfil their transport needs by municipal bike, urban transport, the passenger car they owned, or a car rented based on the car-sharing system. The data show that the majority of MEVO municipal bike users subjected to the survey included regular users who used municipal bikes to meet their transport needs at least several times a week.

**Table 1.** Characteristics of the survey sample ($n = 500$).

| Respondent Characteristics | Mean | SD | Min | Max |
|---|---|---|---|---|
| Age (in years) | 28 | 5.7 | 16 | 59 |
| Household size (persons) | 2.9 | 1.5 | 1 | 7 |
| Household bicycle ownership | 1.2 | 1.4 | 0 | 7 |
| **Respondent Characteristics** | **Characteristic Value (%)** | | | |
| Gender | (Male—48.21%; female—51.56%) | | | |
| Education | (Higher (bachelor's or master's degree)—64.73%; middle school—18.08%; in the course of bachelor or master studies—13.61%; primary education only—3.23%) | | | |
| Social and professional status * | (Working on a contract—78.79%; student—13.83%; pensioner—2.45%; freelancing—6%; unemployed—4%) | | | |
| Driver's license | (Yes—84.38%; No—15.62%) | | | |
| Car ownership | (Yes—65.4%; No—34.6%) | | | |
| Disposable income per month per capita in household | (Below 250 EUR—7.14%; 251–500 EUR—6.02%; 501–750 EUR—26.79%; 751–1000 EUR—25.45%; 1001–1500 EUR—15.63%; above 1500 EUR—18.75%) | | | |
| Main purpose of using MEVO | (Traveling to work—63.84%; traveling to the place of study—9.60%; sport or recreation—6.03%; shopping—14.29%; visiting friends or family—3.79%; other—2.23%) | | | |
| Distance between place of residence and place of work or study | (Below 2 km—11.38%; 2–3 km—30.13%; 4–6 km—22.32%; 7–9 km—16.52%; 10–13 km—6.47%; 14–20 km—8.04%; above 20 km—4.91%) | | | |
| How often do you travel by MEVO bike? | (Every day—10%; several times a week—51%; once a week—6%; several times a month—21%; once a month—3%; less than once a month—9%; first time—0.4%) | | | |

* Multiple answers possible. Source: own elaboration based on the data obtained in the study.

## 4. Results

*4.1. Substitutability of MEVO Municipal Electric Bike Systems against the Other Forms of Urban Transport*

To verify the hypothesis formulated in this article, the respondents' answers to the three following questions were absolutely essential:

- What form of transportation do you replace most frequently with the MEVO municipal bike system?
- If for some reason you would not be able to travel by MEVO bike today (e.g., no bike at the start station or system failure), how would you travel to your destination?
- How did your previous transport behavior change after the launch of the MEVO municipal bike system?

The data presented in Table 2 shows that based on the respondents' answers, travel with the use of municipal electric bikes replaced, to a large extent, the sustainable forms of transportation. In the case of over 50% of the respondents, the municipal bike replaced their travels by public transport. Over 30% of the following respondents abandoned non-motorized forms of transportation (walking and travel by private bicycle) in favor of the municipal bikes. Only 14% of the respondents declared that, for them, the municipal electric bike most frequently replaced the passenger car (including private passenger car, taxi services, car-sharing, and carpooling services). It should also be noted that every fifth surveyed MEVO system user declared that the municipal bike was used instead of walking. In their case, the change in transport behavior, namely walking being partially replaced by travel with the use of municipal electric bikes, should be considered unfavorable for sustainable urban development.

**Table 2.** What does the metropolitan bike replace for you most often? (*n* = 500).

| Mode of Transportation | Percentage of Cases | Mode of Transportation | Percentage of Cases | Mode of Transportation | Percentage of Cases |
|---|---|---|---|---|---|
| passenger car (incl. taxi, car-sharing, and carpooling) | 14.1 | passenger car (incl. taxi, car-sharing, and carpooling) | 14.1 | passenger car (incl. taxi, car-sharing, and carpooling) | 14.1 |
| public railway transport | 18.1 | total public transport | 52.4 | sustainable forms of transportation | 85.9 |
| public transport (bus, trolleybus, and tram) | 34.3 | | | | |
| private bicycle | 12.5 | non-motorized forms of transportation | 33.5 | | |
| walking trips | 21.0 | | | | |

Source: own elaboration based on the data obtained in the study.

Moreover, Table 3 presents data on the respondents' declarations regarding the form of transportation most frequently replaced by municipal electric bikes in their urban travels. However, in this case, the survey covered only the respondents who had a driver's license and a passenger car in their household at the time of the survey. Therefore, the data presented in the table include only those respondents who may truly be in a dilemma over traveling by private passenger car or using other forms of transportation. This is very specific, because people who carry out the significant investment of purchasing a car are then naturally more inclined to use it as the vast part of the costs had already been spent. The results related to people with a driver's license and access to a passenger car were similar to the results obtained for the entire population. Therefore, only for every fifth surveyed respondent who had a real possibility to travel by passenger car was travel by a municipal bike involved in the reduction in car travel.

**Table 3.** What does the metropolitan bike replace for you most often? Answers are from people who have a private passenger car in the household and a driver's license (*n* = 275).

| Mode of Transportation | Percentage of Cases | Mode of Transportation | Percentage of Cases | Mode of Transportation | Percentage of Cases |
|---|---|---|---|---|---|
| passenger car (incl. taxi, car-sharing, and carpooling) | 21.8 | passenger car (incl. taxi, car-sharing, and carpooling) | 21.8 | passenger car (incl. taxi, car-sharing, and carpooling) | 21.8 |
| public railway transport | 17.1 | total public transport | 40.7 | sustainable forms of transportation | 78.2 |

| | | | | |
|---|---|---|---|---|
| public transport (bus, trolleybus, and tram) | 23.6 | | | |
| private bicycle | 14.2 | non-motorized forms of transportation | 37.5 | |
| walking trips | 23.3 | | | |

Source: own elaboration based on the data obtained in the study.

The fact that MEVO municipal electric bikes substituted the sustainable forms of transportation to the largest extent is confirmed by the data presented in Table 4. The data prove that over 80% of the respondents declared that if for some reason they were not able to make their planned travel by the municipal electric bike, they would travel by one of the sustainable forms of transportation. If for some reason travel by municipal bike was impossible, nearly 50% of the respondents would travel by public transport. Less than 20% of the respondents declared that if travel by MEVO municipal bike was impossible, they would travel by passenger car.

**Table 4.** If for some reason you would not be able to travel by a MEVO municipal bike today (e.g., no bike at the start station or system failure), how would you travel to your destination? (*n* = 500)

| Mode of Transportation | Percentage of Cases | Mode of Transportation | Percentage of Cases |
|---|---|---|---|
| by passenger car | 17.4 | by passenger car | 17.4 |
| by public transport | 48.5 | sustainable forms of transportation | 81.0 |
| by private bicycle | 10.4 | | |
| on foot | 21.4 | | |
| in another way | 0.7 | | |
| I would abandon traveling | 1.6 | I would abandon traveling | 1.6 |

Source: own elaboration based on the data obtained in the study.

As concluded above, with regard to the transport policies of cities and their sustainable development and maximization of positive effects resulting from the implementation of municipal bike systems, the people who should travel by municipal bikes most often should be those who substitute the passenger car with the municipal bike most often in their urban travels. Table 5 shows that in the analyzed municipal electric bike system, such a relation was not observed. The largest number of regular users was observed in a group of people for whom the municipal bikes substituted public transport to the largest extent.

**Table 5.** What does the metropolitan bike replace for you most often? This regards the frequency of travel by municipal bikes.

| How Often Do You Travel by MEVO Bike? | | What Does the MEVO Metropolitan Bike Replace for You Most Often? | | | | | |
|---|---|---|---|---|---|---|---|
| | | Total Public Transport (%) | | Passenger Car (Incl. Taxi, Car-Sharing, and Carpooling) (%) | | Non-Motorized Forms of Transportation (%) | |
| every day | | 12.8 | | 9.5 | | 6.7 | |
| several times a week | regular users | 58.3 | 75.4 | 42.9 | 66.7 | 42.3 | 53.0 |
| once a week | | 4.3 | | 14.3 | | 4.0 | |
| once a month | | 4.3 | | 0.0 | | 3.4 | |
| several times a month | occasional users | 14.9 | 24.7 | 23.8 | 33.3 | 29.5 | 47.0 |
| less than once a month | | 5.1 | | 9.5 | | 14.1 | |
| first time | | 0.4 | | 0.0 | | 0.0 | |

Source: own elaboration based on the data obtained in the study.

One of the factors that should positively affect the city residents' willingness to change their mode of urban transportation, swapping a passenger car for a municipal bike, was a low fee for using a bike-sharing system. The data presented in Table 6 show that the monthly cost incurred by the system user who traveled using the system for no longer than 90 min per day totalled EUR 2.3. The running costs of a municipal electric bike system are presented in Table 7.

**Table 6.** Prices of particular tariffs of the MEVO system.

| Tariff | Monthly | Annual | Annual Plus | Minute Fee | 2-Day | 2-Day Plus | 5-Day | 5-Day Plus |
|---|---|---|---|---|---|---|---|---|
| Subscription (EUR) | 2.3 | 23.3 | 35 | 0.023 Pre-1 min | 4.7 | 9.3 | 9.3 | 18.7 |
| Initial fee (one-off) (EUR) | 2.3 | 2.3 | 2.3 | 2.3 | 2.3 | 2.3 | 2.3 | 2.3 |
| Tariff time (days) | 30 | 365 | 365 | Unlimited | 2 | 2 | 5 | 5 |
| Daily time in subscription (min) | 90 | 90 | 120 | - | 300 | 700 | 300 | 700 |
| Rate after exceeding the daily time of rental (EUR per 1 min) | 0.012 | 0.012 | 0.012 | 0.023 | 0.012 | 0.012 | 0.012 | 0.012 |

Source: own elaboration based on [61].

**Table 7.** Monthly costs of urban trips made by particular forms of transportation within the analyzed area (EUR per month).

| Form of Transportation | 30 min/Day | 60 min/Day | 90 min/Day | 120 min/Day |
|---|---|---|---|---|
| MEVO (bike left at the station) | 2.3 | 2.3 | 2.3 | 9.9 |
| MEVO (bike left outside the station) | 13.8 | 13.8 | 13.8 | 21.4 |
| Public transport (ticket of one organizer) | 35.3 | 35.3 | 35.3 | 35.3 |
| Public transport (rail and municipal metropolitan ticket of all organizers) | 54.1 | 54.1 | 54.1 | 54.1 |
| Passenger car | 43.2 | 86.4 | 129.6 | 172.8 |
| Car sharing | 126.5 | 253.0 | 379.5 | 506.0 |

Source: own elaboration based on [61–63].

The data presented in Table 7 show the monthly costs of urban trips within the operational area of the MEVO municipal bike scheme by various forms of transport. The analysis covered the costs of travel by MEVO bikes, public transport (within the area managed by one public transport operator and within the entire metropolitan area), owned passenger cars, and passenger cars within a car-sharing scheme. During the analysis, the following assumptions were applied: a city resident travels 21 days per month (this is the monthly average number of working days in Poland) and makes 2 trips per day. In the case of individual motorized forms of transportation, it was assumed that the trip would be characterized by an average speed of 20 km/h. The data presented in the table show that fulfilling the transport needs with the use of municipal bikes was definitely the least expensive alternative. The fact that municipal bikes, as proven above, substituted passenger cars only to a small extent, even though fulfilling urban transport needs with the use of municipal bikes even involved dozens of times lower costs compared with passenger cars, proved that the cost of travel was not the main factor affecting the change of city residents' transport behavior. This conclusion is confirmed by the data presented in Table 8, presenting the mode of using municipal bikes by the respondents. The data show that most of the respondents used municipal bikes for direct trips to their destinations. Therefore, they left the bikes outside the stations of the MEVO system, which involved the need to incur an additional fee to the amount of EUR 0.5.

Furthermore, the article presented analyses related to non-cost factors affecting the substitutability of municipal electric bikes against passenger cars.

### 4.2. Complementarity of MEVO Municipal Electric Bike Systems against Other Forms of Urban Transportation

As indicated above in the literature review, municipal bike schemes aim, for example, to make it easier for passengers to travel to and from public transport terminals or stations (the so-called first and last mile). Therefore, municipal bike schemes should stimulate the demand for public transport services through increasing their time-based availability, since the total urban travel time in regard to door-to-door systems, thanks to the municipal bikes, should be reduced (referring to people who walked the first and the last mile before introducing the municipal bike scheme). The reduced total travel time in regard to the door-to-door system resulted from the higher average speed of electric bike travel compared with walking trips or trips by traditional bicycles. The average speed of walking trips totaled roughly 4.5 km/h [64], whereas the trips by traditional bicycles amounted to 18.8 km/h. Meanwhile, bikes with electric power-assisted steering (Pedelec) reached 21.9 km/h, and electric bikes (S-pedelec) reached 27.9 km/h [65]. The data show that traveling by a bike equipped with electric power-assisted steering was nearly, on average, five times faster than walking. In order to increase the time-based availability of public transport through offering intermodal travels combined with municipal bikes, it is necessary to ensure proper quality of the bike-sharing systems. In this context, the particularly important features of bike-sharing systems include the number of municipal bikes per city resident or system user (which affects their availability) and the mode of bike relocation in systems with no docking stations (which affects the distance between the user and the closest vacant bike).

The data presented in Table 8 indicate that only every ninth surveyed respondent used a MEVO municipal bike most frequently for travel combined with public transport. The majority of the surveyed respondents used the municipal bike for direct trips to their destinations. Therefore, a MEVO municipal bike was used only to a small extent as a means of transportation complementary to public transport.

**Table 8.** In what circumstances do you use MEVO municipal bikes most often? ($n$ = 500)

| Manner of Using Municipal Electric Bike | Percentage of Cases |
|---|---|
| "I travel directly to my destination, leaving the bike outside a docking station" | 66.2 |
| "I travel to a docking station closest to my destination" | 21.8 |
| "I travel to public transport terminals" | 11.9 |

Source: own elaboration based on the data obtained in the study.

### 4.3. Variables Determining the Probability of Replacing Car Journeys with the MEVO Bicycle

The logistic regression models were then constructed to verify the effect of independent variables on the distribution of users substituting commuting by car with the MEVO bicycle. A sequence of two hundred models were assessed in Statistica 13.1. software with all the possible different combinations of independent variables. The Wald Chi-square test was used to assess the significance of these models. The pseudo R-squared values were calculated to determine the explanatory power of the models, and the Akaike information criterion (AIC) was used to compare the relative quality of the models and to help us investigate if any variables which were omitted could have been included in the model, providing further added value. Logistic regression has been widely used to analyze the determinants of transport behavior and is the generally accepted method in similar research (e.g., [66–69]).

A binary variable showing whether someone substituted a car or sustainable means of transport was chosen as the dependent variable. The respondents declaring that they substituted public transport (road or rail), their private bike, or commuting on foot with the MEVO bike system were aggregated together. Thus, a value of one for the dependent variable indicated that the trip made substituted a car trip. A value of zero meant that it did not or that it substituted another type of trip, including public transport trips. The independent variables were divided into two groups: factors and covariates. The factors included the following:

- Reason behind traveling;
- Distance between the place of residence and the place of work;
- Most common use of the MEVO system;
- Gender;
- Education;
- Socioeconomic status;
- Owning a driver's license;
- Disposable income.

The covariates included the following:

- Year of birth;
- Number of people in the household;
- Number of cars in the household;
- Number of bikes in the household.

The results of logistic regression are presented in Table 9. Only the best model with all the significant variables is presented.

**Table 9.** Logistic regression model (*n* = 500).

| Variable | Odds Ratios |
|---|---|
| Year of birth | −0.071 [0.034] ** |
| Number of people in the household | −0.451 [0.143] *** |
| Number of cars in the household | 0.344 [0.146] *** |
| Distance from home to work (less than 3 km) | −0.891 [0.338] *** |
| Distance from home to work (more than 10 km) | 1.430 [0.595] ** |
| Gender (female) | −0.278 [0.126] * |
| Wald Chi-square test (*p*-value) | 0.028 |
| Pseudo R-squared | 0.323 |
| AIC | 289.492 |

Note: logistic regression coefficients with standard errors are in parentheses (*** $p < 0.01$; ** $p < 0.05$; * $p < 0.1$). The dependent variable is the transport choice flag (1 = car, 0 = sustainable means of transport). Source: own computation in Statistica.

Only a few of all the variables turned out to be statistically significant. These included the age, number of people in the household, number of cars in the household, distance from work, and gender. The results prove that for the analyzed group, the younger participants substituted their cars with MEVO rather than substituting public transport. With each year of age, people were relatively 7% more likely to opt for the substitution of transport modes. Moreover, a higher number of people in a household were correlated with a 45% chance (c.p.) of someone substituting a car with MEVO, possibly due to the lower relative availability of cars within the household as perceived by the respondent. On the other hand, a higher number of cars was correlated with a statistical increase of substituting public transport with MEVO, indicating the well-known fact that car owners tend to use their cars, and the bike-sharing system played a mostly recreational role for them. Each additional car in the household was correlated with a decrease in likelihood

by a further 34%. Moreover, people living far away from their places of work more often substituted public transport with MEVO rather than people living within a 3-km radius of their place of work who, in comparison with the average respondent, commonly substituted a car. In this case, they probably were not that dependent on cars in their commutes and thus could allow themselves to choose MEVO despite its initial problems with availability. Finally, women substituted cars with MEVO 28% more often than men.

Interestingly enough, the reason behind the choice of MEVO systems as a mode of transport was not correlated with the willingness to swap the car for a bike. This might result from the fact that at the time of the survey, the MEVO system was in its early stages, and it was very unlikely to have become someone's main mode of transport in everyday commuting. The willingness to substitute cars was not correlated with the respondent's education, socioeconomic status, or disposable income. This stands in contradiction with the international consensus but goes in line with the fact that the passenger car in Poland is not a luxury in any way. It also plays a key role as some social symbol, crossing the borders of socioeconomic divisions. Further studies should probably include the ages of the owned cars so as to mediate for this effect. Interestingly enough, holding a driver's license was not correlated with the choice, perhaps due to a very small number of people being without a driver's license within the analyzed group.

## 5. Discussion and Conclusions

The analysis presented in the article provides several important implications and conclusions that can be used when planning the implementation of similar electric bike-sharing systems in other metropolitan areas. The survey presented in the article showed that the MEVO municipal electric bike scheme constituted the substitutable form of transportation against collective urban transport to a larger extent than against passenger cars. This means that the operation of a MEVO scheme only limited the use of passenger cars in the city residents' daily trips within the scheme's operational area to a small extent. Consequently, a small degree of car travel was substituted with municipal bike travel, meaning that the MEVO system contributed to a small extent to reducing the external costs generated as a result of fulfilling the city residents' transport needs, therefore contributing only to a small extent to improving the sustainability of cities where the system operated. Moreover, the survey presented in the article proves that the MEVO municipal electric bike system constituted a substitutable form of transportation against walking to a larger extent than against passenger cars. Therefore, as a result of using electric bikes, the system contributed to the increase in external costs of the transport system. Additional costs arose mainly from the need to charge batteries used in the bikes and the redistribution of bikes between stations (with the use of conventional vehicles). The above results mean that the effects of the implementation of the MEVO system met the expectations of policymakers to a very limited extent. This shows that despite the significant advantages that electric bikes have over conventional bikes, this study did not prove that electric bikes could replace more private car trips than conventional bikes. This is particularly important given the significant differences between the cost of implementing and maintaining traditional and electric bike-sharing systems. These differences were so significant that they contributed to the temporary suspension of the MEVO system's operation. The findings of the study partially confirm research being carried out in the already existing body of literature on the subject. More specifically, the results are in line with the results obtained in previous similar studies regarding the strength and direction of substitution between bike-sharing systems and public transport as well as personal cars [2,70,71]. However, there are also studies that show the opposite result, specifically that there is a high or even total rate of substitution of car trips by bike-sharing systems. These studies have been mostly carried out within countries with an already strong culture of active commuting (e.g., Sweden), and the results of our study might be partially contradictory due to general social and cultural factors [72,73].

Moreover, the factors affecting the residents' willingness to substitute car travel with municipal bike travel were analyzed, too. It turned out that the factors increasing the probability to swap the car for a municipal electric bike included gender (women were more willing to abandon car travel), young age, living in a multi-person household (which potentially may adversely affect the possibility to use a passenger car), and living within a small distance (less than 3 km) from the place of work or study.

Taking into account all the results mentioned above, it should be noted that 6 months passed since the launch of the MEVO system when the survey was conducted. Due to the organizational difficulties on the part of the system operator, slightly more than 1200 bikes were used on a regular basis out of the scheduled 4080 municipal electric bikes. This means that the system did not reach its full capacity, which would contribute to the increased availability of bikes and consequently to the increased reliability of this mode of transport perceived by the city residents in urban travel within the system's operational area. It can be assumed that if the number of available municipal bikes increased, the willingness to abandon car travel in favor of bike travel would also increase. It is also possible that the long-term effects of introducing an electric bike-sharing system are different from the short-term effects presented in this article. Another limitation of this study results from its nature as a case study (the sample of participants was not representative and should therefore not be generalized to the wider population). Therefore, it cannot be ruled out that a city's electric bike system operating in different conditions will be characterized by a different degree of substitutability and complementarity against other forms of urban transport. An additional limitation of this study is the fact that only the declarations of the MEVO system users were examined and not their actual behavior. Furthermore, the very nature of the topic indicates that the results might change if the study is carried out in different weather conditions, during different seasons, and also during different days of the week, because mobility patterns, especially regarding various types of active commuting including cycling, change depending on these factors. In the long run, the results might also change based on different transport policies as well as sociocultural aspects, such as the social perception of a personal car.

Recent studies show that most respondents stated that COVID-19 would not affect their intention to use bike-sharing systems [74–76]. Moreover, during the pandemic, many countries promoted the use of bicycles as a safe way to meet transportation needs [77]. This means that despite the collapse in demand for bike-sharing services, there is a good chance that these systems will not lose their relevance [78,79]. It also means that as electric vehicles become more popular, more bike-sharing systems using only electric bikes will be introduced in Poland and Europe. Thus, new opportunities will arise to study the role of electric bike-sharing in urban mobility. Future research may use big data analysis and real behavioral data instead of declarative data. There is also still a need for research into the elements of the urban transport system, the change or improvement of which may lead to a greater shift from car trips to electric city bikes. This is confirmed by the contradictory results of studies of individual conventional and electric bike-sharing systems. We are currently unable to determine to what extent the COVID-19 pandemic will delay the restart of the MEVO system.

**Author Contributions:** M.S.: conceptualization, data curation, methodology, formal analysis, and writing—original draft; A.J.: conceptualization, data curation, formal analysis, project administration, writing—original draft, visualization, and funding acquisition; J.S.: conceptualization, data curation, formal analysis, writing—original draft, and funding acquisition. All authors have read and agreed to the published version of the manuscript.

**Funding:** This research was funded by the Faculty of Economics at the University of Gdansk, project No. 539-3210-B362-19.



**Institutional Review Board Statement:** The study was conducted according to the guidelines of the Declaration of Helsinki, and approved by the Ethics Committee of Faculty of Economics, University of Gdańsk on 01.02.2019

**Informed Consent Statement:** Informed consent was obtained from all subjects involved in the study

**Data Availability Statement:** Dataset is available at https://ekonom.ug.edu.pl/pp/download.php?OpenFile=34895

**Conflicts of Interest:** The authors declare that they have no known competing financial interests or personal relationships that could have appeared to influence the work reported in this paper.

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
