# Peer review of "Substitutability and Complementarity of Municipal Electric Bike Sharing Systems against Other Forms of Urban Transport"

_applsci, doi:10.3390/app11156702_

Round 1

Reviewer 1 Report

This paper uses a sample of 500 e-bike bikeshare riders in Poland to estimate mode replacement of e-bikes compared to cars and other modes (6 months after the launch of a limited e-bikesharing system). The authors find that most ebike riders report replacing non-automobile trips.

I have several comments:

First, I am not sure why electric bikeshare bikes have to substitute driving more than other modes to be sustainable. Sustainability is a large concept include access, equity etc. I think the authors focus on CO2 emissions or something, but not on sustainability in general. The authors should define sustainability more clearly.

Second, in terms of mode replacement: what is the mode share of trips for the trip distance categories captured by the paper? For example in table 3: If only 14% of these trips are made by car then there would be a proportional trip replacement for various modes.

Third, and importantly, the paper overall and in particular the results of the regression include causal language which has to be removed. All you observe here are correlations.

Fourth, the discussion of the regression results in confusing. This needs to be clarified. Sometimes the authors refer to public transport, but they indicated that all non-car modes were combined. I think the dependent variable is 1 if bikeshare substituted for car and 0 if bikeshare substituted for other modes.

Fifth, the final regression includes on significant results, but produces contradictory findings. Maybe the regression is plagued by omitted variables bias?

Sixth, the paper is confusing because it tends to present too much. I think some of the many tables and the related discussion should be deleted. The paper needs more focus.

Other items:

Abstract:

I think it should say (test the hypothesis) and not (verified)

I think the abstract assumes municipally owned bikesharing systems. This does not have to be the case.

Review of literature: I think the section on the history of bikesharing can be cut significantly. Just cite other papers like Shaheen, DeMaio, etc.

Lines 260-270: were these bikeshare stations close to public transport?

Line 383: the costs of cars of course work differently. There is a huge investment up-front. Once you have the car, you will drive with it because you already spent so much money one it.

Author Response

We want to express our significant thanks for the review and all the comments, which we believed have helped us improve the paper in an appropriate fashion. We’ve changed significant parts of the text and also provide answers to the comments below.

This paper uses a sample of 500 e-bike bikeshare riders in Poland to estimate mode replacement of e-bikes compared to cars and other modes (6 months after the launch of a limited e-bikesharing system). The authors find that most ebike riders report replacing non-automobile trips.

I have several comments:

First, I am not sure why electric bikeshare bikes have to substitute driving more than other modes to be sustainable. Sustainability is a large concept include access, equity etc. I think the authors focus on CO2 emissions or something, but not on sustainability in general. The authors should define sustainability more clearly.

This was based on the assumption that using personnel cars is considered to be less sustainable (among others in terms of CO2 emissions) than using public transport. If a bike substitutes public transport than the overall effect in terms of sustainability is much less significant than if it substitutes car. We agree that it has not been formulated clearly enough and have added information accordingly at the end of section 2.2.

Second, in terms of mode replacement: what is the mode share of trips for the trip distance categories captured by the paper? For example in table 3: If only 14% of these trips are made by car then there would be a proportional trip replacement for various modes.

The distribution is fairly proportional, with a higher share of participants commuting by walking for shorter distances, naturally. We’ve looked at this but decided not to provide the cross analysis due to the fact, that it didn’t provide any further insight into the problem and, as also observed in the sixth point below, we didn’t want to further increase the size of the paper. Naturally, we can include it, along with a short description, if deemed necessary. We appreciate further insight in this respect

Third, and importantly, the paper overall and in particular the results of the regression include causal language which has to be removed. All you observe here are correlations.

We entirely agree, we’ve corrected this all along the paper.

Fourth, the discussion of the regression results in confusing. This needs to be clarified. Sometimes the authors refer to public transport, but they indicated that all non-car modes were combined. I think the dependent variable is 1 if bikeshare substituted for car and 0 if bikeshare substituted for other modes.

We entirely agree, we’ve corrected this all along the paper.

Fifth, the final regression includes on significant results, but produces contradictory findings. Maybe the regression is plagued by omitted variables bias?

We’ve decided to make the choice regarding the variables in the model based on Akaike’s Information Criterion thus omitting some of the variables. The finding which is the most contradictory one is that (as specified in last paragraphs of the results section) the willingness to substitute cars wasn’t either correlated with the respondent’s education, socio-economic status or disposable income. This stands in contradiction with the international consensus but goes in line with the fact that the passenger car in Poland is not a luxury in any way and very often very cheap cars are purchased not warranting any significant investments. We hope, that after the rewriting, the results section is now acceptable.

Sixth, the paper is confusing because it tends to present too much. I think some of the many tables and the related discussion should be deleted. The paper needs more focus.

We’ve combined some of the data in the material and methods section into one table and shortened parts of the literature review. We believe that the rest of the data is significant in terms of the findings but are happy to further decrease the size of the article, if necessary.

Other items:

Abstract:

I think it should say (test the hypothesis) and not (verified)

This has been rephrased

I think the abstract assumes municipally owned bikesharing systems. This does not have to be the case.

It does but we’ve concentrated on those and MEVO was also municipally owned.

Review of literature: I think the section on the history of bikesharing can be cut significantly. Just cite other papers like Shaheen, DeMaio, etc.

This section has been shortened significantly

Lines 260-270: were these bikeshare stations close to public transport?

They have been. Appropriate information has been added.

Line 383: the costs of cars of course work differently. There is a huge investment up-front. Once you have the car, you will drive with it because you already spent so much money one it.

We agree, we’ve added appropriate information.

Reviewer 2 Report

The authors analyzed responses to a survey administered to users of an electric bike sharing system in order to evaluate the relationship of complementarity or substitution with sustainable transport means. In particular, statistical analysis and logistic regression were applied, pointing out that the current electric systems do not replace many trips on cars. The paper is well organized, the aims and conclusions are clearly stated. Some few comments below.

The authors should consider to add the term “bike sharing” in the title, in order to highlight the type of service they are dealing with.

I suggest the authors to add some sentences describing the contributions of their work respect to the existing literature.

Page 4, figure 1: I suggest to replace “Cyckling” with “Cycling”

Sources in tables can be omitted (except for table 7), since they contain results from the analysis which were clearly carried out by the authors.

Page 12-13, Table 10: I suggest the authors to add odds ratios for each independent variable, since in page 13 they reported some comments about these values.

Author Response

We want to express our significant thanks for the review and all the comments, which we believed have helped us improve the paper in an appropriate fashion. We’ve changed significant parts of the text and also provide answers to the comments below.

The authors analyzed responses to a survey administered to users of an electric bike sharing system in order to evaluate the relationship of complementarity or substitution with sustainable transport means. In particular, statistical analysis and logistic regression were applied, pointing out that the current electric systems do not replace many trips on cars. The paper is well organized, the aims and conclusions are clearly stated. Some few comments below.

Thank you very much for those kind words

The authors should consider to add the term “bike sharing” in the title, in order to highlight the type of service they are dealing with.

We’ve changed the title accordingly

I suggest the authors to add some sentences describing the contributions of their work respect to the existing literature.

Thoughts on this have been added in the discussion section

Page 4, figure 1: I suggest to replace “Cyckling” with “Cycling”

This has been changed

Sources in tables can be omitted (except for table 7), since they contain results from the analysis which were clearly carried out by the authors.

This has been done according to the journal’s indications but we may delete them if necessary.

Page 12-13, Table 10: I suggest the authors to add odds ratios for each independent variable, since in page 13 they reported some comments about these values.

There has been a general title “values” which might have been confusing and has now been changes into odds ratios.

Reviewer 3 Report

It is advisable to make the topic more immediate by making the title of the article more attractive.
In the introductory section, it is advisable to include more literature on the evolution of bike sharing and to diversify the different classes of users that could benefit from using this transport service. 
1) Nikiforiadis, A., Ayfantopoulou, G., & Stamelou, A. (2020). Assessing the impact of COVID-19 on bike-sharing usage: The case of Thessaloniki, Greece. Sustainability, 12(19), 8215.
2) Torrisi, V., Ignaccolo, M., Inturri, G., Tesoriere, G., & Campisi, T. (2021). Exploring the factors affecting bike-sharing demand: evidence from student perceptions, usage patterns and adoption barriers. Transportation Research Procedia, 52, 573-580.
3)Boufidis, N., Nikiforiadis, A., Chrysostomou, K., & Aifadopoulou, G. (2020). Development of a station-level demand prediction and visualization tool to support bike-sharing systems' operators. Transportation Research Procedia, 47, 51-58.
It is advisable to include a picture/map that can identify the area of the analysed transport service.
It is advisable to include more details on how the data was found and whether it was influenced by COVID-19.

It is advisable to specify the limitations of the research conducted by specifying which factors may change the possible results when the sites change (e.g. weather conditions, social aspects, etc.).

Author Response

We want to express our significant thanks for the review and all the comments, which we believed have helped us improve the paper in an appropriate fashion. We’ve changed significant parts of the text and also provide answers to the comments below.

It is advisable to make the topic more immediate by making the title of the article more attractive.

We’ve changed the title to include the bikesharing name so as to make it more relevant for the readers.

In the introductory section, it is advisable to include more literature on the evolution of bike sharing and to diversify the different classes of users that could benefit from using this transport service. 
1) Nikiforiadis, A., Ayfantopoulou, G., & Stamelou, A. (2020). Assessing the impact of COVID-19 on bike-sharing usage: The case of Thessaloniki, Greece. Sustainability, 12(19), 8215.
2) Torrisi, V., Ignaccolo, M., Inturri, G., Tesoriere, G., & Campisi, T. (2021). Exploring the factors affecting bike-sharing demand: evidence from student perceptions, usage patterns and adoption barriers. Transportation Research Procedia, 52, 573-580.
3)Boufidis, N., Nikiforiadis, A., Chrysostomou, K., & Aifadopoulou, G. (2020). Development of a station-level demand prediction and visualization tool to support bike-sharing systems' operators. Transportation Research Procedia, 47, 51-58.

We’ve included the articles and believe they significantly increased our knowledge of the topic.

It is advisable to include a picture/map that can identify the area of the analysed transport service.

Unfortunately, it’s the journal’s policy not to include any copyrighted material and we do not have any access to a free of charge map of the area.

It is advisable to include more details on how the data was found and whether it was influenced by COVID-19.

The data has been collected pre-COVID – we’ve added appropriate information.

It is advisable to specify the limitations of the research conducted by specifying which factors may change the possible results when the sites change (e.g. weather conditions, social aspects, etc.).

We’ve added additional information, especially in the discussion section.

Round 2

Reviewer 1 Report

Thank you for adressing my comments wherever possible.

Author Response

Thank you for all your help and insight

Reviewer 3 Report

It is necessary to reformulate figure 1 to make it more readable.
It is necessary to standardise the formatting of the tables.
The manuscript still has some grammatical errors in the text.
Once these corrections have been made, the manuscript will be eligible for publication. 

Author Response

We've made all the corrections indicated. We appreciate your insight and thank you for your time.